# A Project-Scheduling and Resource Management Heuristic Algorithm in the Construction of Combined Cycle Power Plant Projects

**Shakib Zohrehvandi** 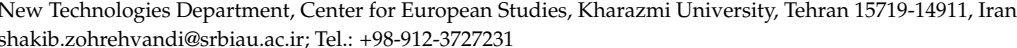

New Technologies Department, Center for European Studies, Kharazmi University, Tehran 15719-14911, Iran; shakib.zohrehvandi@srbiau.ac.ir; Tel.: +98-912-3727231

**Abstract:** Given the growing number of development projects, proper project planning and management are crucial. The purpose of this paper is to introduce a heuristic algorithm for scheduling a power plant project construction and project resource management to determine the size of project buffers and feeding buffers. This algorithm consists of three steps: 1. estimating the duration of project activities; 2. determining the size of the project buffer and feeding buffers; and 3. simulating the mentioned algorithm, which will be explained below. Innovations of this research are as follows: estimating the exact duration of project activities by using a heuristic algorithm, in addition to determining the buffer size; calculating both project buffer and feeding buffers; and applying the algorithm to implement an ACC used in combined cycle power plant projects as a numerical example. In order to evaluate the proposed algorithm, inputs from this project were run through several algorithms recently presented. The results showed that a suitable amount of buffers can be allocated for projects using this algorithm.

**Keywords:** construction planning; project scheduling; heuristic algorithm; resource management; project buffer management; combined cycle power plant project

## 1. Introduction

Most development projects, whether implemented by the public sector or assigned to a contractor, are not completed within the planned time and cost. Extending the time and increasing the budget of the project, perhaps several times, are widespread consequences. These problems are mostly due to poor planning and inadequate project control by project executives. The main task of the project planning and control system is to prepare, collect, record, and store information on the various stages of the project life cycle process; classify and analyze the information; and prepare the required reports for the project manager. The purpose of this system is to direct the project according to the set schedule and budget; reach the goals and final products of the project; and store the obtained information for use in future projects [1]. Zohrehvandi et al. [2] introduced a reconfigurable model that combines a schedule model and a queuing system M/M/m/K to reduce the duration of a wind turbine construction project closure phase and to reduce project documentation waiting time in a queue. Zohrehvandi et al. [3] applied appropriate planning for the implementation of a combined cycle power plant project in the closing phase with minimum deviation. One optimization method to increase the project's scheduling stability is to create a buffer (safety time) by using the critical chain method to deal with time changes in the project. Three types of buffers are used in project scheduling: project buffer (PB), feeding buffer (FB), and resource buffer (RB). The project buffer is placed at the end of the project critical path to maintain the project's delivery date. Feeding buffers are added to the paths that connect to the critical chain so that possible delays will not affect the critical chain. As for resource buffers, they ensure that the project resources are ready when a critical activity requires them [4].

Hall [5] conducted extensive research on project management research opportunities for the next 10 years. In his research, one of the areas of project management, which was considered to have a high potential for future research, was project scheduling and project buffer management. Buffer sizing is one of the most important steps in project buffer management. It can be considered the most important measure in the implementation of critical chain scheduling, because if the allocated buffers are too short, we will need to reschedule them many times until the end of the project, and if they are too long, all scheduling concepts will be violated. Therefore, it is crucial to use a suitable buffer sizing method. One of the most important aspects of managing a project in the critical chain management method is determining the buffer's size. Buffer size determination should be based on the project specifications, project status, and the risks involved so that the project can be optimally protected from possible delays. In addition to the above, the following should also be considered in determining buffer size [6]:

- Type of project activities;
- Number and size of project activities;
- Complexities of and relationships between activities;
- The number of resources available to the activities and the resources required;
- The number of uncertainties in the project environment.

To make buffer sizing more accurate, in this research, the indices of resource accessibility, resource flexibility, resource quality, and resource sustainability of activities are examined. The purpose of this paper is to introduce a heuristic algorithm for scheduling power plant project construction and the project resource management for determining the size of project buffers and feeding buffers. For a more accurate management of buffers, a combination of the Delphi method and Program Evaluation and Review Technique (PERT) was used to determine the exact duration of activities. The algorithm mentioned in this research is presented by a numerical example in the implementation of an ACC, which is used in power plant projects. The algorithm consists of three steps: 1. estimating the exact duration of project activities through the integration of Delphi and PERT methods; 2. determining the size of project buffer and feeding buffers; and 3. simulating the algorithm by using the Monte Carlo method. The Monte Carlo method uses the repetition of simulation to understand the behavior of a phenomenon. The tendency to use the Monte Carlo method increases when it is impossible or unjustified to calculate the exact answer with the help of deterministic algorithms [7].

One of the major limitations with projects is that they are not completed according to schedule. Uncertainty always exists at the heart of real-world project scheduling problems. The required information was obtained through interviews with experts and elites. The information required for using the proposed algorithm has been gathered by holding meetings with the project experts.

## 2. Literature Review

Today, the development of methods resulting in the success of projects in the phases of planning, implementation, and completion is receiving considerable attention. Competition in the field of project management is constantly increasing, and the project team always needs to complete projects on time and within budget. To achieve this, project managers need to plan and control the project under uncertain conditions and with resource constraints (RCPSP). Liu et al. [8] presented an optimization model for project planning processes under uncertainty and with resource constraints. One of the methods used in project planning and control to reduce the project execution time and make it more realistic is buffer management in project implementation, which is derived from the critical chain project management method.

The critical chain project management approach was first introduced by Goldratt [4] to improve the traditional project management approach by using a new mechanism for managing uncertainties. Since the publication of Goldratt's theory, several studies have

been conducted by some researchers, including Newbold [9], Leach [10], Tukel et al. [6], Woeppel [11], and Rabbani et al. [12].

The theory of constraints and its direct application in project management under the title of Critical Chain/Buffer Management (CC/BM) has been a popular and effective project management approach developed by Goldratt [13]. Hammad et al. [14] presented a new framework for estimating, allocating, and managing planning probabilities using the theory of constraints and obtained value. The theory of constraints can serve as a new approach to ensure better control over project execution using buffer management. Newbold [9] improved Goldratt's critical chain management theory and introduced the RSEM method, which is considered one of the traditional buffer management methods. The main disadvantage of this technique is that resource constraints are not taken into account in buffer sizing. Tukel et al. [6] introduced newer methods than the critical chain management theory of Goldratt and Newbold [3,8]. They introduced APD and APRT methods, which are traditional buffer management methods. Vanhoucke [15] examined traditional buffer sizing methods and how to obtain them and compared the results by using an example.

Zohrehvandi and Khalilzadeh [16] presented an efficient algorithm for project buffer sizing by taking failure mode and effects analysis (FMEA) into account and reached a more realistic schedule. The proposed algorithm was implemented on a real wind farm construction project. Many researchers have been able to develop and improve traditional project buffer management methods; for example, Zhang et al. [17] proposed a new approach in using the CCPM method by considering an information-based relationship between project activities. Zhang et al. [18] introduced a fuzzy approach to project buffer sizing. Hu et al. [19] presented six prioritization indices for selecting an optimal chain when more than one chain is possible. Then, they examined four production plans for rescheduling. Sarkar et al. [20] developed a project critical chain management framework for the effective implementation of construction projects.

Poshdar et al. [21] investigated a probability-based buffer allocation method in which buffer size was determined by project planners based on preferences. Beşikci et al. [22] introduced a multiproject planning environment that included several projects with specific dates. They presented three scheduling problems to explore this multi-project environment. In their research, they integrated this multiproject environment as one model and presented it as a resource portfolio problem. They also used a genetic algorithm as a solution. Rueda-Velasco et al. [23] proposed an algorithm for scheduling multiple projects with dynamic resource allocation. Research has been conducted in the field of buffer sizing. Rahman et al. [24] proposed an algorithm based on a genetic algorithm to solve a resource-constrained project planning problem. They implemented the proposed algorithm in the critical path of the project. It was a heuristic algorithm based on the critical path. In this regard, Zhang et al. [25] proposed a buffer sizing method based on resource tightness to better reflect the relationships between activities and improve the accuracy of project buffer sizing.

Bakry et al. [26] proposed a buffer sizing algorithm for optimal planning of construction projects under uncertainty conditions and used fuzzy set theory to model the uncertainties associated with the input parameters. Zarghami et al. [27] introduced a new buffer sizing method by modeling the CCPM method through a project resource reliability analysis. Zhao et al. [28] proposed a buffer sizing method that considered resource interference in critical chain project management. Zohrehvandi et al. [29] introduced an efficient project buffer and resource management (PBRM) model for project resource leveling and project buffer sizing and controlling of project buffer consumption of a wind power plant project to achieve a more realistic project duration. Moreover, Zohrehvandi et al. [30] introduced a heuristic algorithm to determine the sizes of project buffer and feeding buffers as well as dynamically control buffer consumption, named Fuzzy Overlapping Buffer Management Algorithm (FOBMA). Recently, Zohrehvandi and Soltani [31] discussed the state-of-the-art models and methods for project buffer management and time optimization of construction projects and manufacturing industries.

Saravanan et al. [32] tried to find the best possible function of integrated power plants to obtain the most favorable solution relative to the planning and scheduling problem. Leyman and Vanhoucke [33] examined the resource-constrained project scheduling problem. They introduced a scheduling method to improve the net present value (NPV) of the project. Almeida et al. [34] explored one of the newest methods of resource-constrained project scheduling. They used a parallel schedule in this method. Bruni et al. [35] raised a resource-constrained project scheduling problem. Kadri and Boctor [36] introduced a resource-constrained project scheduling problem with resource transfer times (RCPSPTT). Bevilacqua et al. [37] examined a real problem consisting of multi-objective optimization of planning a project's activities by taking resource constraints and prioritization into account. They used the CCPM method in this study. Cheraghi et al. [38] presents a multiproject scheduling and resource management (MPSRM) model that includes an M/M/c/n queue system, a p-hub median model, a parallel machine scheduling and a hub location problem solution method. They aims to design a project network and then sequence raw materials delivery to hub factories. The results suggested that the proposed model significantly reduces project transportation costs.

The PERT method was first introduced by the US Navy for a large and complex submarine project. The PERT method is the most extensive technique for project planning, scheduling, and controlling and a method for project evaluation and review [39]. One assumption is the Beta distribution with a three-point estimate: optimistic ($a$), most probable ($m$), and pessimistic ($b$), and the mean is $\mu = (a + (4 * m) + b)/6$. The use of several distributions with other parameter estimates has been proposed [40]. The Delphi method is used to gather the experts' specialized views about the phenomenon under study. In other words, this method is used to achieve group consensus in specialized fields. The Delphi method is a structural process for gathering knowledge by using a series of open-ended questionnaires with controlled feedback to reach a consensus. One of the main advantages of Delphi is that it can result in an agreement even when there is no evidence in a specific field and there is uncertainty [41].

According to the literature review, both Delphi and PERT methods have not been applied together with project buffer management algorithms in the previous studies. The above literature shows that the proposed algorithm has not yet been used in previous studies and no similar algorithm has been presented so far. The literature review in this study is categorized according to the topics related to the methodology presented in this study, as shown in Table 1.

**Table 1.** Comparison of previous research with this research.

| Author/Year | Research Subjects | | | | |
| --- | --- | --- | --- | --- | --- |
| | Project Scheduling and Resource Management | PERT Technique | Delphi Technique | Buffer Sizing Techniques | Monte Carlo Method |
| Hazır 2015; Zohrehvandi et al., 2019; Zohrehvandi et al., 2017; Beşikci et al., 2015; Rueda-Velasco et al., 2017; Rahman et al., 2020; Zohrehvandi et al., 2020a; Saravanan et al., 2018; Leyman and Vanhoucke 2015; Almeida et al., 2016; Bruni et al., 2017; Kadri and Boctor 2018; Bevilacqua et al., 2015; Cheraghi et al., 2022; Liu et al., 2020 | ✓ | | | | |

**Table 1.** *Cont.*

| Author/Year | Research Subjects | | | | |
|---|---|---|---|---|---|
| | Project Scheduling and Resource Management | PERT Technique | Delphi Technique | Buffer Sizing Techniques | Monte Carlo Method |
| Goldratt 1997; Hall 2015; Tukel et al., 2006; Newbold 1998; Leach 2005; Woeppel 2006; and Rabbani et al., 2007; Goldratt 1984; Hammad et al., 2018; Vanhoucke 2016; Zohrehvandi and Khalilzadeh 2019; Zhang et al., 2015; Zhang et al., 2017; Hu et al., 2019; Sarkar et al., 2021; Poshdar et al., 2016; Zhang et al., 2016; Bakry et al., 2016; Zarghami et al., 2019; Zhao et al., 2020; Zohrehvandi et al., 2020b; Zohrehvandi and Soltani 2022 | | | | ✓ | |
| Hohmann et al., 2018 | | | ✓ | | |
| Alekseeva et al., 2018 | | | | | ✓ |
| Salas-Morera et al., 2018; Hajdu and Bokor 2016 | | ✓✓ | | | |
| This Research | ✓ | ✓ | ✓ | ✓ | ✓ |

The above table shows that the method presented in this research has not been used in previous research until now, which shows the article's novelty. The method proposed in this research uses all five issues in the table above and presents a new model.

## 3. Methodology

The algorithm presented in this research is schematically shown in Figure 1. In general, this algorithm consists of three steps: 1. estimating project activities duration; 2. determining the size of the project buffer and feeding buffers; and 3. simulating the mentioned algorithm, which will be explained below. The algorithm in this study is presented as a numerical example in the implementation of an ACC used in power plant projects.

According to the algorithm presented in this research (Figure 1), the PERT method was used to determine and estimate the appropriate duration of project activities. Once the resources required for the project activities have been determined by holding meetings with the project experts, the critical path of the project was determined, and the schedule of the entire project was finalized. Then, in order to determine the exact duration of the project activities, the optimistic, pessimistic, and most likely estimates related to the PERT method were determined through meetings with project experts. Finally, after determining the mentioned numbers, the exact duration of the project activities was specified.

In order to better estimate the duration of project activities in this research, the Delphi method was implemented in parallel with the PERT method. To implement this method, meetings were held with project experts and elites, and participants in the meetings were asked about the project activities' durations. In the next step, the average value of comments collected was considered as the result of the Delphi method. Finally, the results of the above methods were combined with equal weight ratios, and the final and exact durations of project activities were estimated.

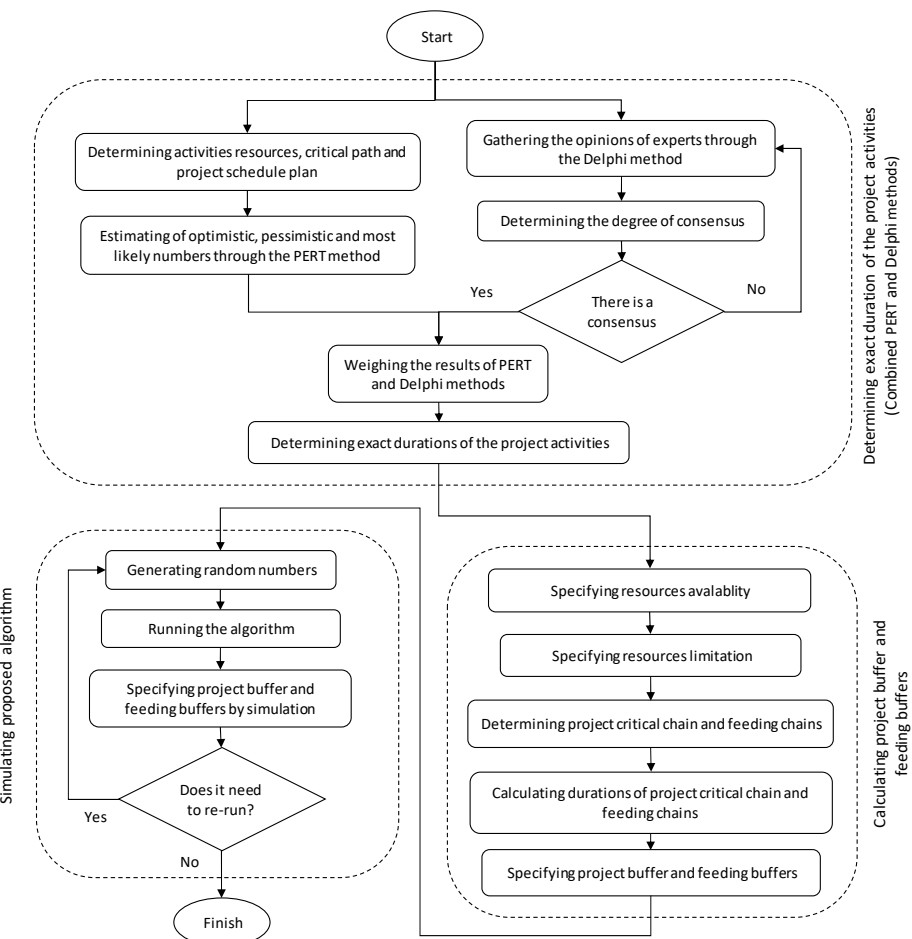

**Figure 1.** The proposed buffer management algorithm.

After determining the exact duration of project activities, the resource accessibility index, resource flexibility index, resource sustainability index, and resource quality index of activities were specified by holding meetings with experts and elites related to the project to determine the amount of project resource constraints. Then, the number of resources available in the project was calculated, and the amount of project resource constraints was determined. At this stage, the medium resource constraints of each activity, as well as the highest level of resource constraints of each activity in the project were determined. Finally, the planned size of the project buffer and feeding buffers were estimated by setting the medium resource constraints and the highest level of resource constraints.

Indices, parameters, and variables used in the proposed algorithm of this research are as follows:

**Indices:**

Resource index: $(q = 1 \ldots Q)$;

Project activities index: $(u = 1 \ldots U)$;

Index of project critical activities: $(c = 1 \ldots C)$;

Feeding chain index: $(f = 1 \ldots F)$;

Index of project activities duration (the result of Delphi method): $(d = 1 \ldots D)$;

**Parameters**:

$D_u$: Initial duration of Activity $u$;

$D_d$: Duration of Activity $u$ (the result of Delphi method);

$O_u$: Optimistic duration of Activity $u$;

$ML_u$: Most likely duration of Activity $u$;

$P_u$: Pessimistic duration of Activity $u$;

$A_{qu}$: Accessibility index of Resource $q$ in Activity $u$;

$F_{qu}$: Flexibility index of Resource $q$ in Activity $u$;
$Q_{qu}$: Quality index of Resource $q$ in Activity $u$;
$S_{qu}$: Sustainability index of Resource $q$ in Activity $u$;
$Rpr(u;q)$: Accessibility of Resource $q$ to Activity $u$;
$V_u$: Time ratio of Activity $u$ in the critical chain;
**Variables:**
$r(u;q)$ : Amount of Resource $q$ for Activity $u$;
$D_{Pu}$: Exact duration of Activity $u$ in the project using the PERT method;
$D_{du}$: Exact duration of Activity $u$ in the project using the Delphi method;
$D_{pdu}$: Exact duration of Activity $u$ in the project using PERT and Delphi integrated method;
$RC(u;q)$: Amount of Resource $q$ constraints for Activity $u$;
$RCM_u$: Medium resource constraints for Activity $u$;
$RCH_u$: The highest level of resource constraints for activity $u$;
$Y$: Project buffer compatibility index;
$PB$: Size of project planned buffer.

As explained in the previous sections, the algorithm presented in this research consists of three steps.

*3.1. Determining the Exact Duration of the Project Activities*

Once the required resources for the project activities ($r(u;q)$) have been determined by holding meetings with project experts, the critical path of the project was determined and the overall project schedule was finalized. $r(u;q)$ is the amount of Resource $q$ for Activity $u$. To determine the exact duration of project activities, the Optimistic, Most Likely, and Pessimistic numbers denoted in Equation (1) as $O_u$, $ML_u$, and $P_u$, respectively, were determined by holding meetings with project experts. After determining the mentioned numbers, the exact duration of the project activities was determined by the PERT method, which is shown in Equation (1) as $D_{Pu}$.

$$D_{Pu} = (O_u + (4 * ML_u) + P_u)/6 \tag{1}$$

The Delphi method was also implemented in parallel with the PERT method. To implement this method, meetings were held with project experts and elites, and the participants were consulted about project activity durations. Then, the average number of collected comments was considered as the result of the Delphi method. In Equation (2), $D_{du}$ is the exact duration of activity $u$ in the project, which was calculated using the Delphi method. Moreover, $D_d$ is the duration of activity $u$, which was determined by conducting a survey on project experts, and $D$ is the number of comments raised in the meeting.

$$D_{du} = \left( \sum_{d=1}^{D} D_{\hat{d}} \right)/D \tag{2}$$

Finally, the results of the PERT and Delphi methods were combined with equal weight ratios, and the final and exact durations of project activities were estimated. In Equation (3), $D_{pdu}$ is the exact duration of activity $u$ in the project, which was obtained by combining Delphi and PERT methods.

$$D_{pdu} = (D_{Pu} * 0.5) + (D_{du} * 0.5) \tag{3}$$

*3.2. Calculating Project Buffer and Feeding Buffers*

In this section, the size of the project buffer and feeding buffers was calculated. In Equation (4), $A_{qu}$, $F_{qu}$, $Q_{qu}$, and $S_{qu}$ are, respectively, the indices of accessibility, flexibility, quality, and sustainability of Activity $u$ resource in the project and are considered important and decisive in determining the amount of resource constraint $RC(u;q)$. The size of the project buffer and feeding buffers were calculated using an approach proposed by Zhang et al. [18]. In this research, to calculate the amount of activities resource constraints more accurately, the accessibility and flexibility indices of activity resources were used. In

addition to determining the project's buffer size, the feeding buffers were also calculated. In Equation (4), $Rpr(u;q)$ denotes the accessibility of Resource $q$ for Activity $u$.

$$RC(u;q) = \frac{r(u;q)}{Rpr(u;q) * Q_{qu} * A_{qu} * F_{qu} * S_{qu}} \tag{4}$$

Chang et al. [42] studied the allocation of the required resources for reconstruction projects. They achieved a comprehensive resource planning process for activities. They also examined activities resources accessibility and identified the factors that play a role in determining them. After conducting the research, it was found that the activities resource accessibility $(A_{qu})$ is one of the most important items in resource planning. The medium resource constraint of Activity u and the highest level of resource constraint of the activity were calculated according to Equations (5) and (6), respectively. In Equation (5), the number of resources of Activity $u$ is represented by $Q$. Equations (5) to (8) were adopted from a study by Zohrehvandi et al. (2020).

$$RCM_u = (\sum_{q=1}^{Q} RC(u;q))/Q \tag{5}$$

$$RCH_u = \underset{q=1...n}{Max}(RC(u;q)) \tag{6}$$

In Equation (7), the compatibility index of the project buffer is represented by $Y$ and is obtained by the following equation.

$$Y = (1 + RCM_u) * (1 + RCH_u) \tag{7}$$

In Equation (8), the planned project buffer is denoted by $PB$. Furthermore, $u = C$ indicates that all calculated activities were critical. Finally, $V_u$ is the time ratio of Activity $u$ in the critical chain.

$$PB = \sqrt{\sum_{u=1}^{U}(Y * V_u)} \tag{8}$$

### 3.3. Simulating Proposed Algorithm

In order simulate the proposed algorithm, it was been coded and run in MATLAB software, and the Monte Carlo method was used to generate random numbers to solve the algorithm.

## 4. Numerical Example

The Gantt chart in Figure 2 shows the initial schedule of an Air-Cooled Condenser (ACC) project in full. This schedule has been used to run similar projects in the past. This schedule includes the phases of Engineering, Manufacturing, procurement, and Installation, which were designed in MSP software. According to this schedule, the Engineering phase, the Manufacturing and procurement phase, and the Installation phase were planned to last 150, 169, and 380 days, respectively. Carrying out all phases, which is the entire execution of an ACC, was scheduled to last 435 days. This schedule consists of 44 rows including milestones, activities, and summary activities. In this schedule, activities are marked with blue and red horizontal charts, and the red activities are considered critical. Critical activities are activities with zero float time. In this schedule, activities 4 to 6, 10, 21 to 24, 26 to 29, and 32 to 43 are critical and form the critical path of this project.

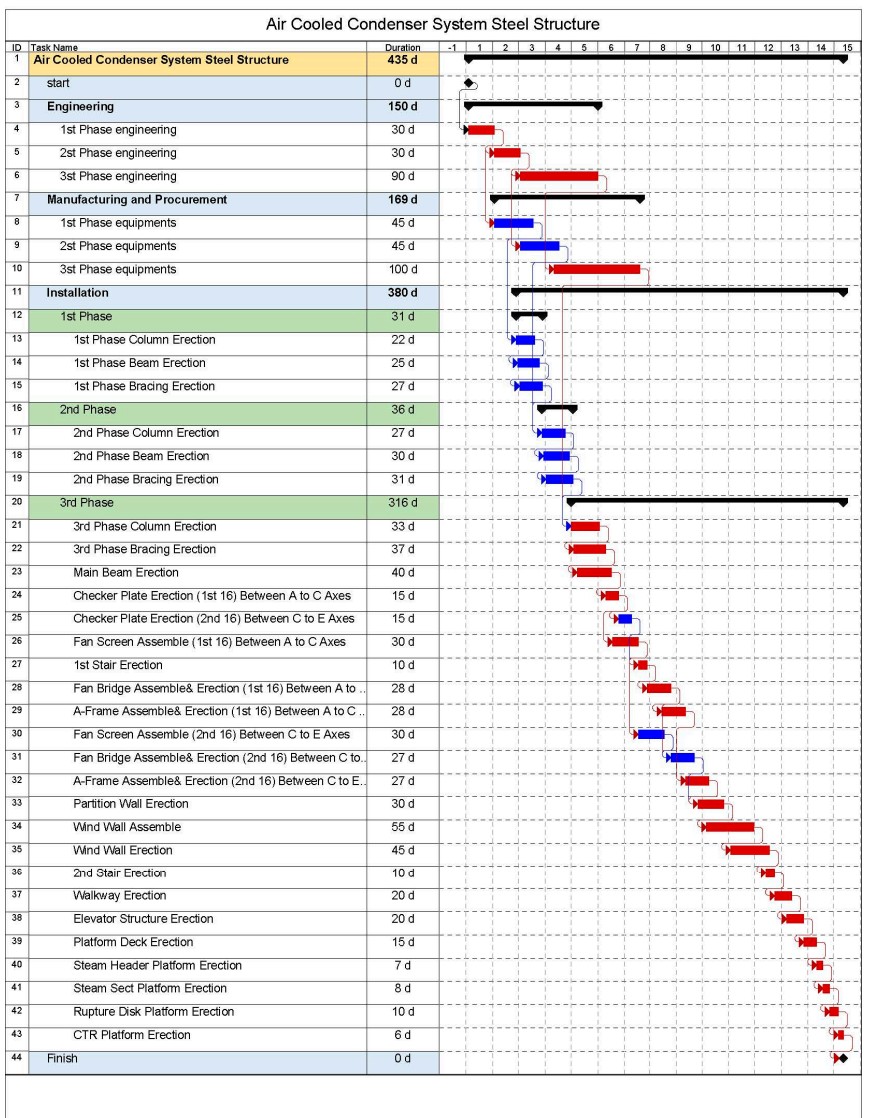

**Figure 2.** The initial schedule plan of an ACC.

Table 2 shows the project schedule information in more detail. This table provides information on Predecessor and Successor activities.

**Table 2.** Basic information about the project plan.

| Task Name | Duration | Predecessors | Successors |
|---|---|---|---|
| Air Cooled Condenser System Steel Structure | 435 d | | |
| start | 0 d | | 4 |
| Engineering | 150 d | | |
| 1st Phase engineering | 30 d | 2 | 8, 5 |
| 2nd Phase engineering | 30 d | 4 | 9, 6 |
| 3rd Phase engineering | 90 d | 5 | 10FS-51 d |
| Manufacturing and Procurement | 169 d | | |
| 1st Phase equipment | 45 d | 4 | 13FS-20 d |
| 2st Phase equipment | 45 d | 5 | 17FS-20 d |
| 3rd Phase equipment | 100 d | 6FS-51 d | 21FS-80 d |

**Table 2.** *Cont.*

| Task Name | Duration | Predecessors | Successors |
|---|---|---|---|
| Installation | 380 d | | |
| 1st Phase | 31 d | | |
| 1st Phase Column Erection | 22 d | 8FS-20 d | 14FS-20 d |
| 1st Phase Beam Erection | 25 d | 13FS-20 d | 15FS-23 d |
| 1st Phase Bracing Erection | 27 d | 14FS-23 d | 17FS-5 d |
| 2nd Phase | 36 d | | |
| 2nd Phase Column Erection | 27 d | 15FS-5 d, 9FS-20 d | 18FS-25 d |
| 2nd Phase Beam Erection | 30 d | 17FS-25 d | 19FS-27 d |
| 2nd Phase Bracing Erection | 31 d | 18FS-27 d | 21FS-5 d |
| 3rd Phase | 316 d | | |
| 3rd Phase Column Erection | 33 d | 19FS-5 d, 10FS-80 d | 22FS-30 d |
| 3rd Phase Bracing Erection | 37 d | 21FS-30 d | 23FS-33 d |
| Main Beam Erection | 40 d | 22FS-33 d | 24FS-7 d |
| Checker Plate Erection (1st 16) Between A and C Axes | 15 d | 23FS-7 d | 26FS-7 d, 25 |
| Checker Plate Erection (2nd 16) Between C and E Axes | 15 d | 24 | 27 |
| Fan Screen Assemble (1st 16) Between A and C Axes | 30 d | 24FS-7 d | 27, 30 |
| 1st Stair Erection | 10 d | 26, 25 | 28 |
| Fan Bridge Assemble and Erection (1st 16) Between A and … | 28 d | 27 | 29FS-11 d, 31 |
| A-Frame Assembly and Erection (1st 16) Between A and C … | 28 d | 28FS-11 d | 32 |
| Fan Screen Assemble (2nd 16) Between C and E Axes | 30 d | 26 | 31 |
| Fan Bridge Assemble and Erection (2nd 16) Between C and… | 27 d | 28, 30 | 33 |
| A-Frame Assembly and Erection (2nd 16) Between C and E … | 27 d | 29 | 33FS-13 d |
| Partition Wall Erection | 30 d | 32FS-13 d, 31 | 34FS-20 d |
| Wind Wall Assemble | 55 d | 33FS-20 d | 35FS-27 d |
| Wind Wall Erection | 45 d | 34FS-27 d | 36FS-4 d |
| 2nd Stair Erection | 10 d | 35FS-4 d | 37 |
| Walkway Erection | 20 d | 36 | 38FS-6 d |
| Elevator Structure Erection | 20 d | 37FS-6 d | 39 |
| Platform Deck Erection | 15 d | 38 | 40 |
| Steam Header Platform Erection | 7 d | 39 | 41 |
| Steam Sect Platform Erection | 8 d | 40 | 42 |
| Rupture Disk Platform Erection | 10 d | 41 | 43 |
| CTR Platform Erection | 6 d | 42 | 44 |
| Finish | 0 d | 43 | |

*4.1. Determining the Exact Duration of the Project Activities*

The number of resources required for the activities was determined b interviews with experts and elites, as shown in Table 3. According to the table, there are three types of resources to perform the activities in the project, which are represented by q1 to q3. However, all three types of resources may not be used for each activity. For example, in Activity 6 (3rd Phase engineering), two types of resources were used, and in Activity 17 (2nd Phase Column Erection), all three types of resources were used.

**Table 3.** The appropriate amount of project activity resources.

| Activity Number ($u$) | Task Name | $r(u,q)$ | | |
|---|---|---|---|---|
| 1 | Air Cooled Condenser System Steel Structure | q1 | q2 | q3 |
| 2 | start | | | |
| 3 | Engineering | | | |
| 4 | 1st Phase engineering | 4.7 | 3.0 | 4.2 |
| 5 | 2nd Phase engineering | - | 4.2 | 6.9 |
| 6 | 3rd Phase engineering | 2.9 | 3.0 | - |
| 7 | Manufacturing and Procurement | | | |
| 8 | 1st Phase equipment | 4.8 | - | 5.8 |
| 9 | 2st Phase equipment | 5.1 | 4.9 | 4.2 |
| 10 | 3rd Phase equipment | 3.4 | 3.0 | - |
| 11 | Installation | | | |
| 12 | 1st Phase | | | |
| 13 | 1st Phase Column Erection | 5.0 | - | 2.7 |
| 14 | 1st Phase Beam Erection | 3.4 | 3.0 | - |
| 15 | 1st Phase Bracing Erection | 4.4 | 2.9 | 3.4 |
| 16 | 2nd Phase | | | |
| 17 | 2nd Phase Column Erection | 5.9 | 4.4 | 4.1 |
| 18 | 2nd Phase Beam Erection | 3.9 | - | 5.0 |
| 19 | 2nd Phase Bracing Erection | 3.0 | 4.0 | 3.0 |
| 20 | 3rd Phase | | | |
| 21 | 3rd Phase Column Erection | 4.7 | 3.0 | 4.2 |
| 22 | 3rd Phase Bracing Erection | - | 4.2 | 6.9 |
| 23 | Main Beam Erection | 2.9 | 3.7 | - |
| 24 | Checker Plate Erection (1st 16) Between A and C Axes | 5.9 | 4.4 | 4.1 |
| 25 | Checker Plate Erection (2nd 16) Between C and E Axes | 4.8 | - | 5.8 |
| 26 | Fan Screen Assemble (1st 16) Between A and C Axes | 5.1 | 4.9 | 4.2 |
| 27 | 1st Stair Erection | 7.6 | 8.5 | - |
| 28 | Fan Bridge Assemble and Erection (1st 16) Between A and . . . | 3.4 | 3.0 | - |
| 29 | A-Frame Assembly and Erection (1st 16) Between A and C . . . | 4.4 | 2.9 | 3.4 |

**Table 3.** *Cont*.

| Activity Number (*u*) | Task Name | r(*u,q*) | | |
|---|---|---|---|---|
| 30 | Fan Screen Assemble (2nd 16) Between C and E Axes | 4.0 | - | 4.0 |
| 31 | Fan Bridge Assemble and Erection (2nd 16) Between C and... | 3.4 | 3.0 | - |
| 32 | A-Frame Assembly and Erection (2nd 16) Between C and E . . . | 4.4 | 2.9 | 3.4 |
| 33 | Partition Wall Erection | 7.6 | 8.5 | - |
| 34 | Wind Wall Assemble | 5.9 | 4.4 | 4.1 |
| 35 | Wind Wall Erection | 3.9 | - | 4.9 |
| 36 | 2nd Stair Erection | 5.0 | 4.0 | 3.7 |
| 37 | Walkway Erection | 4.0 | 4.0 | 3.7 |
| 38 | Elevator Structure Erection | 7.6 | 4.0 | - |
| 39 | Platform Deck Erection | 3.4 | 4.0 | - |
| 40 | Steam Header Platform Erection | 4.4 | 2.9 | 3.4 |
| 41 | Steam Sect Platform Erection | 5.9 | 4.4 | 4.1 |
| 42 | Rupture Disk Platform Erection | 4.8 | - | 5.8 |
| 43 | CTR Platform Erection | 5.1 | 4.9 | 4.2 |
| 44 | Finish | | | |

After the required resources for the project activities were identified, a combination of PERT and Delphi methods was used to determine the exact duration of project activities. Both methods were implemented in parallel. In the PERT method, Optimistic, Pessimistic, and Most likely numbers were determined by conducting meetings with project experts. Then, the exact duration of the activities was obtained by using the PERT method, as presented in Table 4.

**Table 4.** Exact times for project activities by PERT method.

| Task Name | Optimistic | Most Likely | Pessimistic | PERT Expected Duration |
|---|---|---|---|---|
| Air Cooled Condenser System Steel Structure | | | | 421 |
| start | | | | 0 |
| Engineering | | | | 144 |
| 1st Phase engineering | 24 | 29 | 31 | 29 |
| 2nd Phase engineering | 25 | 28 | 33 | 28 |
| 3rd Phase engineering | 80 | 87 | 95 | 87 |
| Manufacturing and Procurement | | | | 159 |
| 1st Phase equipment | | | | 45 |
| 2st Phase equipment | | | | 45 |
| 3rd Phase equipment | 88 | 95 | 104 | 95 |
| Installation | | | | 367 |
| 1st Phase | | | | 31 |
| 1st Phase Column Erection | | | | 22 |

**Table 4.** *Cont.*

| Task Name | | Optimistic | Most Likely | Pessimistic | PERT Expected Duration |
|---|---|---|---|---|---|
| | 1st Phase Beam Erection | | | | 25 |
| | 1st Phase Bracing Erection | | | | 27 |
| 2nd Phase | | | | | 36 |
| | 2nd Phase Column Erection | | | | 27 |
| | 2nd Phase Beam Erection | | | | 30 |
| | 2nd Phase Bracing Erection | | | | 31 |
| 3rd Phase | | | | | 307 |
| | 3rd Phase Column Erection | 29 | 34 | 36 | 34 |
| | 3rd Phase Bracing Erection | 30 | 35 | 38 | 35 |
| | Main Beam Erection | 33 | 38 | 43 | 38 |
| Axes | Checker Plate Erection (1st 16) Between A and C | 10 | 13 | 15 | 13 |
| Axes | Checker Plate Erection (2nd 16) Between C and E | | | | 15 |
| Axes | Fan Screen Assemble (1st 16) Between A and C | 27 | 31 | 35 | 31 |
| | 1st Stair Erection | 10 | 12 | 14 | 12 |
| Between A and . . . | Fan Bridge Assemble and Erection (1st 16) | 21 | 25 | 26 | 25 |
| A and C . . . | A-Frame Assembly and Erection (1st 16) Between | 20 | 24 | 26 | 24 |
| Axes | Fan Screen Assemble (2nd 16) Between C and E | | | | 30 |
| Between C and... | Fan Bridge Assemble and Erection (2nd 16) | | | | 27 |
| Between C and E . . . | A-Frame Assembly and Erection (2nd 16) Between | 23 | 29 | 37 | 29 |
| | Partition Wall Erection | 25 | 31 | 38 | 31 |
| | Wind Wall Assemble | 46 | 50 | 55 | 50 |
| | Wind Wall Erection | 40 | 43 | 45 | 43 |
| | 2nd Stair Erection | 9 | 11 | 13 | 11 |
| | Walkway Erection | 19 | 23 | 26 | 23 |
| | Elevator Structure Erection | 18 | 21 | 25 | 21 |
| | Platform Deck Erection | 14 | 16 | 19 | 16 |
| | Steam Header Platform Erection | 4 | 6 | 8 | 6 |
| | Steam Sect Platform Erection | 5 | 7 | 9 | 7 |
| | Rupture Disk Platform Erection | 8 | 10 | 11 | 10 |
| | CTR Platform Erection | 5 | 6 | 8 | 6 |
| Finish | | | | | 0 |

According to the schedule, the total project time was reduced from 435 days to 421 days. The Delphi method was implemented in tandem with the PERT method. To perform this, meetings were held with experts and project elites, and participants were asked

to opine on the project activities durations. The mean values of the collected comments are given in Table 5.

**Table 5.** Exact times for project activities by Delphi method.

| Activity Number ($u$) | Task Name | Delphi Expected Duration |
|---|---|---|
| 1 | Air Cooled Condenser System Steel Structure | 414 |
| 2 | start | |
| 3 | Engineering | 143 |
| 4 | 1st Phase engineering | 27 |
| 5 | 2nd Phase engineering | 31 |
| 6 | 3rd Phase engineering | 85 |
| 7 | Manufacturing and Procurement | 161 |
| 8 | 1st Phase equipment | 43 |
| 9 | 2st Phase equipment | 47 |
| 10 | 3rd Phase equipment | 96 |
| 11 | Installation | 364 |
| 12 | 1st Phase | 30 |
| 13 | 1st Phase Column Erection | 20 |
| 14 | 1st Phase Beam Erection | 27 |
| 15 | 1st Phase Bracing Erection | 26 |
| 16 | 2nd Phase | 32 |
| 17 | 2nd Phase Column Erection | 25 |
| 18 | 2nd Phase Beam Erection | 27 |
| 19 | 2nd Phase Bracing Erection | 32 |
| 20 | 3rd Phase | 300 |
| 21 | 3rd Phase Column Erection | 31 |
| 22 | 3rd Phase Bracing Erection | 33 |
| 23 | Main Beam Erection | 42 |
| 24 | Checker Plate Erection (1st 16) Between A and C Axes | 12 |
| 25 | Checker Plate Erection (2nd 16) Between C and E Axes | 16 |
| 26 | Fan Screen Assemble (1st 16) Between A and C Axes | 28 |
| 27 | 1st Stair Erection | 10 |
| 28 | Fan Bridge Assemble and Erection (1st 16) Between A and . . . | 26 |
| 29 | A-Frame Assembly and Erection (1st 16) Between A and C . . . | 25 |
| 30 | Fan Screen Assemble (2nd 16) Between C and E Axes | 30 |
| 31 | Fan Bridge Assemble and Erection (2nd 16) Between C and... | 28 |
| 32 | A-Frame Assembly and Erection (2nd 16) Between C and E . . . | 25 |
| 33 | Partition Wall Erection | 33 |
| 34 | Wind Wall Assemble | 53 |
| 35 | Wind Wall Erection | 44 |
| 36 | 2nd Stair Erection | 9 |
| 37 | Walkway Erection | 22 |

**Table 5.** *Cont.*

| Activity Number (*u*) | Task Name | Delphi Expected Duration |
|---|---|---|
| 38 | Elevator Structure Erection | 19 |
| 39 | Platform Deck Erection | 17 |
| 40 | Steam Header Platform Erection | 8 |
| 41 | Steam Sect Platform Erection | 6 |
| 42 | Rupture Disk Platform Erection | 9 |
| 43 | CTR Platform Erection | 7 |
| 44 | Finish | |

According to the Delphi method results, the total project duration obtained was 414 days. As previously explained, Delphi and PERT methods were combined in this study to better estimate the project activities' durations. The total duration of the project using the PERT and Delphi methods was 421 and 414 days, respectively. After combining these two methods and assigning equal weights to each of them, the accurate estimate of the total project duration was 417 days, 18 days less than the initial project duration (435 days).

*4.2. Calculating Project Buffer and Feeding Buffers*

After estimating the exact duration of the project activities, research entered the buffer sizing phase. Figure 3 shows the project activity network. The critical path is also marked in red. There were 24 critical activities in the critical path of the project. At this stage, the size of the project buffer and feeding buffers was determined. Then, the critical chain of the project and the feeding chains were identified.

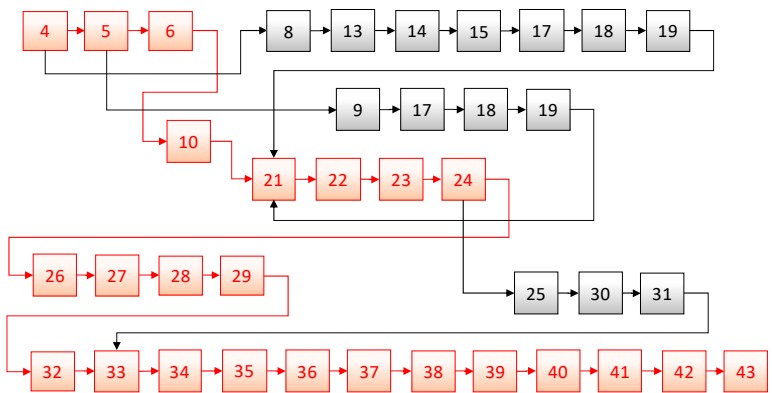

**Figure 3.** The project activities network.

At this stage, the resource accessibility index ($A_{qu}$) and the resource sustainability index of project activities ($S_{qu}$) were determined by holding meetings with project experts and elites to obtain the amount of project resource constraints. Furthermore, to determine the amount of project resource constraints, the flexibility index ($F_{qu}$) and the quality index of activities resources ($Q_{qu}$) were also determined by conducting meetings. The amount of project resource constraints ($RC(u; q)$) was determined after calculating the available resources in the project ($Rpr(u; q)$).

At this stage, the medium resource constraint ($RCM_u$) and the highest level of resource constraint of each project activity ($RCH_u$) were determined. The planned size of project buffer and feeding buffers was determined by setting the medium resource constraint and the highest level of resource constraint.

Finally, according to Table 6, the planned buffer size of the project and the size of the feeding buffers were obtained. The size of the project buffer was 35 days, and the size

of feeding buffers 1, 2, and 3 was determined to be 20.8, 11.1, and 9.5 days, respectively. Moreover, the length of the project critical chain (without buffer) was 417 days, and the lengths of feeding chains 1, 2, and 3 were 203.5, 132, and 73 days, respectively. The 'Plan to buffer' column also contains the sum of the planned durations of the chains and the planned buffer durations. The length of the project critical chain was 452 days taking buffers into account, and the lengths of feeding chains 1, 2, and 3 taking account of buffers were 224.3, 143.1, and 82.5 days, respectively.

**Table 6.** Planned project buffer and feeding buffers.

| Items | Chains | Chains Duration | | Planned Buffers Duration |
|---|---|---|---|---|
| | | Plan | Plan with Buffer | |
| 1 | 4-5-6-10-21-22-23-24-26-27-28-29-32-33-34-35-36-37-38-39-40-41-42-43 | 417.0 | 452.0 | 35.0 |
| 2 | 4-8-13-14-15-17-18-19-21 | 203.5 | 224.3 | 20.8 |
| 3 | 5-9-17-18-19-21 | 132.0 | 143.1 | 11.1 |
| 4 | 24-25-30-31-33 | 73.0 | 82.5 | 9.5 |

Figure 4 presents the critical chain of the project and feeding chains 1, 2, and 3. In this figure, the location of the project buffer and feeding buffers are shown. The project buffer was placed at the end of the project critical path, and the feeding buffers were placed at the end of the chains entering the critical path.

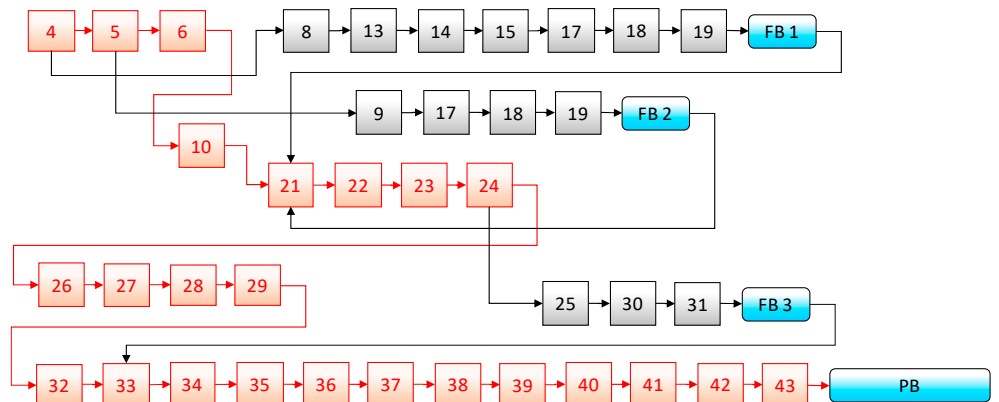

**Figure 4.** The project activities network with buffers.

### 4.3. Simulating Proposed Algorithm

In order to simulate the heuristic algorithm presented in this research, the algorithm was coded and run in MATLAB software. Then, the Monte Carlo method was used to generate random numbers to solve the algorithm. The simulation results of this research were performed 1000 times by the Monte Carlo method, the results of which can be observed in Figure 5, Figure 6, Figure 7, Figure 8, Figure 9. Figure 5 shows the planned duration of the project buffer for the number of times the algorithm was executed, which is related to the variable of the medium resource constraint ($RCM_u$) in performing the 1st Phase engineering Activity.

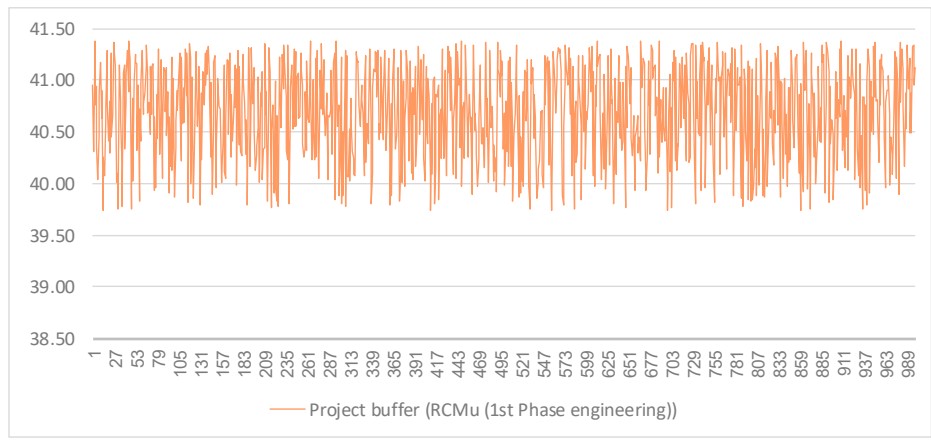

**Figure 5.** Simulating of planned project buffers (1st Phase engineering).

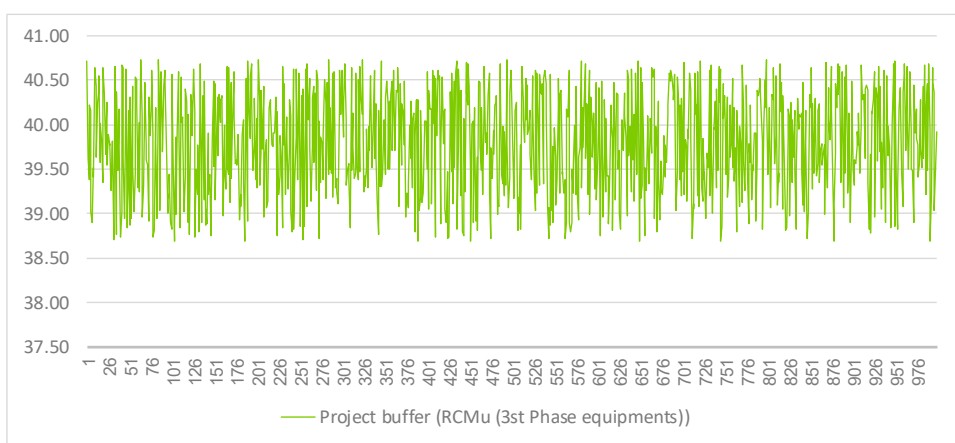

**Figure 6.** Simulating of planned project buffers (3rd Phase equipment).

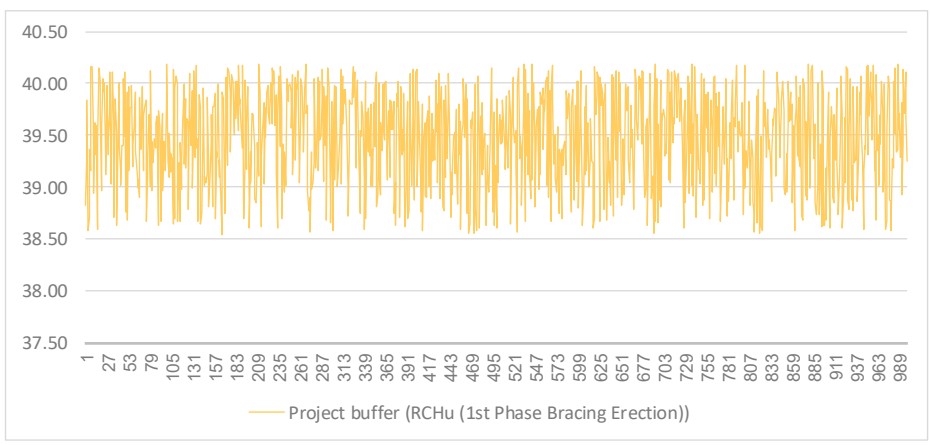

**Figure 7.** Simulating of planned project buffers (1st Phase bracing erection).

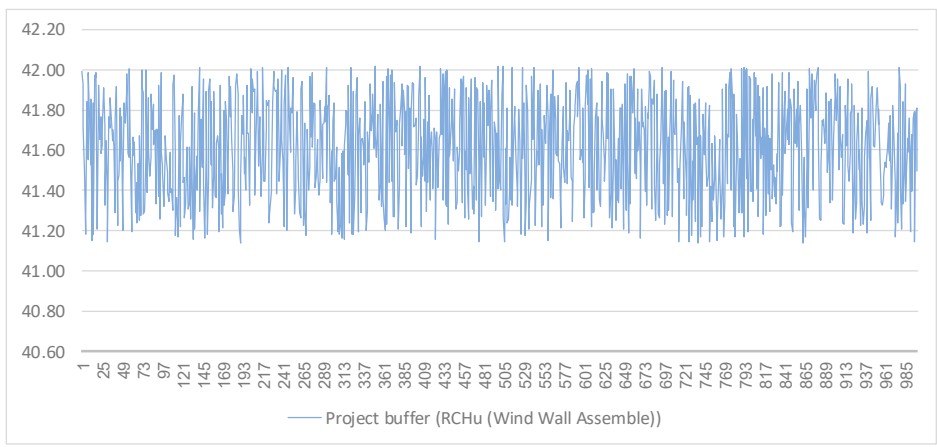

**Figure 8.** Simulating of planned project buffers (wind wall assemble).

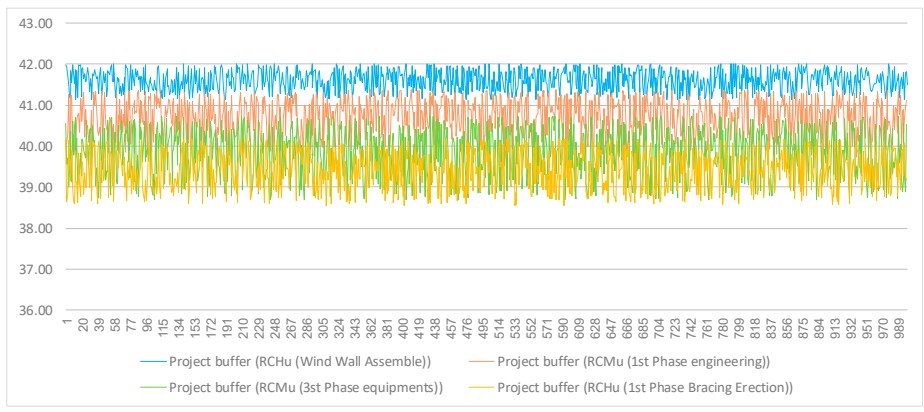

**Figure 9.** Simulating of planned project buffers (1st Phase engineering, 3rd Phase equipment, 1st Phase Bracing erection, and wind wall assemble).

Figure 6 shows the planned duration of the project buffer for the number of times the presented heuristic algorithm was executed, which is related to the variable of the medium resource constraint ($RCM_u$) in performing the 3rd Phase equipment Activity.

Figure 7 shows the planned duration of the project buffer for the number of times the proposed heuristic algorithm was executed, which is related to the variable of the highest level of resource constraint ($RCH_u$) in performing the 1st Phase bracing erection Activity.

Figure 8 shows the planned duration of the project buffer for the number of times the presented heuristic algorithm was executed, which is related to the variable of the highest level of resource constraint ($RCH_u$) in performing wind wall assemble Activity.

The result of simulation through the Monte Carlo method is roughly shown in Figure 9. This diagram compares the two variables of the medium resource constraint ($RCM_u$) and the highest level of resource constraint ($RCH_u$) in the Activities 1st Phase engineering, 3rd Phase equipment, 1st Phase activities bracing erection, and wind wall assemble in terms of the planned duration of the project buffer.

The results of this simulation showed that by making modifications in the variables $RCM_u$ and $RCH_u$ in the mentioned activities, the planned duration of the project buffer was affected in a certain interval, which is not much different from the determined amount of project buffers in this research.

## 5. Findings and Results

In order to evaluate the proposed algorithm, the inputs from this project were run through several algorithms recently presented, and the obtained results are presented in Table 7 for the sake of comparison.

**Table 7.** Comparison of the proposed algorithm with that of the previous algorithms.

| Items | Models/Authors | Planned Project Buffer (Days) |
|---|---|---|
| 1 | Buffer sizing model/Zhang et al. (2017) | 37 |
| 2 | APRT-FMEA buffer sizing/Zohrehvandi and Khalilzadeh (2019) | 40 |
| 3 | This research | 35 |

According to the results shown above, the planned project buffer using the proposed algorithm was 13% less than the APRT-FMEA buffer sizing algorithm used by Zohrehvandi and Khalilzadeh [16]. Moreover, the results of implementing the proposed algorithm showed that the planned project buffer was 6% lower compared to the buffer sizing algorithm presented by Zhang et al. [18].

A combination of the PERT and Delphi methods was used to determine the exact duration of project activities in this research. In the PERT method, optimistic, pessimistic, and most likely numbers were determined by conducting meetings with project experts, and project duration was reduced from 435 days to 421 days as a result. The Delphi method was performed in tandem with the PERT method. To implement this method, meetings were held with experts and project elites, and participants were consulted about the project activities' durations. Project duration was reduced from 435 days to 414 days by using the Delphi method. Then, to better estimate the duration of project activities, the Delphi and PERT methods were combined. By assigning equal weights to each of them, the accurate estimate of the total project time was obtained to be 417 days, which was 18 days less than the initial project time (435 days).

In the next step, the planned sizes of the project buffer and feeding buffers were determined. Accordingly, the size of the project buffer was 35 days, and the size of feeding buffers 1, 2, and 3 were 20.8, 11.1, and 9.5 days, respectively. Furthermore, the length of the critical chain of the project including the buffer duration was 452 days, and the lengths of feeding chains 1, 2, and 3 including the buffer duration were 224.3, 143.1, and 82.5 days, respectively. In order to simulate the proposed algorithm, the Monte Carlo method was used to generate random numbers to solve the algorithm. The simulation results of this research were performed 1000 times by using the Monte Carlo method. Additionally, in this simulation, a comparison was made regarding the planned project buffer times among variables $RCM_u$ and $RCH_u$ in the following Activities: 1st Phase engineering, 3rd Phase equipment, 1st Phase bracing erection, and wind wall assemble.

The algorithm proposed in this research has not been implemented in practice, but by using the Monte Carlo method, virtual information has been obtained randomly and the implementation of this algorithm has been simulated.

## 6. Conclusions

The purpose of this paper was to introduce a heuristic algorithm for scheduling a power plant project construction and project resource management for determining the size of project buffers and feeding buffers. In order to have a more accurate management of buffers, a combination of PERT and Delphi methods was used to determine the exact time of activities. Innovations of this research study are as follows: estimating the exact duration of project activities by the integration of Delphi and PERT methods, in addition to determining the buffer size; calculating project buffer and feeding buffers; and applying the algorithm to implement an ACC used in combined cycle power plant projects as a numerical example.

In the first stage, after the required resources for the project activities were identified by holding meetings with the project experts, the critical path of the project was determined, and the entire project schedule was finalized. Then, to determine the exact duration of the project activities, a combination of the PERT and Delphi methods was used. In the

second stage, by conducting meetings with project experts and elites, the indices of resource accessibility, resource flexibility, resource sustainability, and resource quality of the activities were determined to identify project resource constraints. Then, the available resources in the project were calculated; finally, the number of resource constraints was determined. At this stage, the medium resource constraints and the highest level of resource constraint of each activity in the project were determined. Finally, the planned size of the project buffer and feeding buffers was determined by setting the medium resource constraint and the highest level of resource constraint. In the third stage, to simulate the proposed algorithm, the algorithm was coded and run in MATLAB software, and the Monte Carlo method was used to generate random numbers to solve the algorithm. The results of this simulation showed that by making changes in variables $RCM_u$ and $RCH_u$ in the Activities (1st Phase engineering, 3rd Phase equipment, 1st Phase bracing erection, and wind wall assemble), the planned duration of the project buffer was affected at a certain point in time, which is not much different from the set amount of project buffer in this research study. Therefore, it can be concluded that, by using this algorithm, a suitable amount of buffers can be considered for projects, which can result in increasing the productivity of projects and completing them in a shorter time period than the scheduled one.

This research can be beneficial for researchers, project managers, industry owners, and all those who deal with projects. This research can be implemented in all projects in addition to power plant projects. Moreover, it is possible to implement this proposed algorithm on real projects as case studies. The consumption of buffers has not been discussed in this study, which can be considered as future studies by other researchers. Considering the positive results of the implementation of this algorithm in this project, the proposed algorithm can be used in future research by other researchers.

**Funding:** This research did not receive any specific grant from funding agencies in the public, commercial, or not-for-profit sectors.

**Institutional Review Board Statement:** Not applicable.

**Informed Consent Statement:** Not applicable.

**Data Availability Statement:** Data is contained within the article.

**Acknowledgments:** I am grateful to all of those with whom I have had the pleasure to work during this and other related projects.

**Conflicts of Interest:** The author declares no conflict of interest.

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
