# Peer review of "A Project-Scheduling and Resource Management Heuristic Algorithm in the Construction of Combined Cycle Power Plant Projects"

_computers, doi:10.3390/computers11020023_

Round 1

Reviewer 1 Report

The paper presents an interesting combination of various methods (Delphi, PERT, Critical Chain) into one scheduling algorithm. Project buffers are determined based on various resource availabilities, which is a correct approach to buffer determination. The algorithm has been applied to a real world project. It is a pity there is no information about the realisation of the schedule. Was it actually implemented? What was the buffer usage during the implementation? The authors should make a reference to this.

Author Response

The word file is attached

Reviewer 2 Report

The paper entitled “A project-scheduling and resource management heuristic algorithm in construction of combined cycle power plant projects” deals with a very interesting topic, and it included interesting ideas. The approach followed looks useful and the results are promising.

However, I have the following comments that hopefully help the authors improve their paper:

  • I suggest that the authors add a research method diagram. This will provide a snapshot of the research steps followed and will help the reader in a clearer understanding of the paper.
  • In relation to literature review, it would be better if authors can have a table comparing the closely related works on various dimensions and clearly showing the contribution of the paper.
  • What are the limitations of the study in terms of the proposed method, data used, approaches, and/or analysis?
  • How can you apply in practice and real life this approach? Is it feasible? Authors should clearly state the limitations of the proposed method in practical applications.
  • The main contribution of this paper should be compared with other similar empirical studies.
  • The main contribution of this paper compared to the available literature needs to be better outlined. The authors should convince the readers of this journal, that their contribution is so important. These issues deserve a deeper discussion: What are the managerial implications from this work? What are the implications for theory and practice?
  • As usual a final thorough proof-reading is recommended.

I encourage the author to think along those questions and to develop this work further along those lines.

Author Response

The word file is attached

Round 2

Reviewer 2 Report

The manuscript has significantly improved as compared to the previous version. Indeed, the authors tried to improve it, and the main weaknesses are solved. 

I am also satisfied with the responses and explanations given by the authors to my comments.

Thus, in my opinion, the manuscript is recommendable for publication.